# Functional Brain Network Topology Discriminates between Patients with Minimally Conscious State and Unresponsive Wakefulness Syndrome

**DOI:** 10.3390/jcm8030306

**Published:** 2019-03-05

**Authors:** Alberto Cacciola, Antonino Naro, Demetrio Milardi, Alessia Bramanti, Leonardo Malatacca, Maurizio Spitaleri, Antonino Leo, Alessandro Muscoloni, Carlo Vittorio Cannistraci, Placido Bramanti, Rocco Salvatore Calabrò, Giuseppe Pio Anastasi

**Affiliations:** 1Department of Biomedical, Dental Sciences and Morphological and Functional Images, University of Messina, 98125 Messina, Italy; dmilardi@unime.it (D.M.); dptbiosciences@gmail.com (G.P.A.); 2Brain Bio-Inspired Computing (BBC) Lab, IRCCS Centro Neurolesi Bonino Pulejo, 98124 Messina, Italy; g.naro11@alice.it (A.N.); alessia.bramanti@irccsme.it (A.B.); leonardo.malatacca@irccsme.it (L.M.); maurizio.spitaleri@irccsme.it (M.S.); antonino.leo@irccsme.it (A.L.); kalokagathos.agon@gmail.com (C.V.C.); dino.brama@gmail.com (P.B.); 3Biomedical Cybernetics Group, Biotechnology Center (BIOTEC), Center for Molecular and Cellular Bioengineering (CMCB), Center for Systems Biology Dresden (CSBD), Department of Physics, Technische Universität Dresden, Tatzberg 47/49, D-01307 Dresden, Germany; alessandro.muscoloni@gmail.com

**Keywords:** functional connectome, consciousness, fronto-parietal connectivity, network analysis, local-community-paradigm

## Abstract

Consciousness arises from the functional interaction of multiple brain structures and their ability to integrate different complex patterns of internal communication. Although several studies demonstrated that the fronto-parietal and functional default mode networks play a key role in conscious processes, it is still not clear which topological network measures (that quantifies different features of whole-brain functional network organization) are altered in patients with disorders of consciousness. Herein, we investigate the functional connectivity of unresponsive wakefulness syndrome (UWS) and minimally conscious state (MCS) patients from a topological network perspective, by using resting-state EEG recording. Network-based statistical analysis reveals a subnetwork of decreased functional connectivity in UWS compared to in the MCS patients, mainly involving the interhemispheric fronto-parietal connectivity patterns. Network topological analysis reveals increased values of local-community-paradigm correlation, as well as higher clustering coefficient and local efficiency in UWS patients compared to in MCS patients. At the nodal level, the UWS patients showed altered functional topology in several limbic and temporo-parieto-occipital regions. Taken together, our results highlight (i) the involvement of the interhemispheric fronto-parietal functional connectivity in the pathophysiology of consciousness disorders and (ii) an aberrant connectome organization both at the network topology level and at the nodal level in UWS patients compared to in the MCS patients.

## 1. Introduction

The human connectome is a comprehensive description of neural elements and their reciprocal connections reflecting the complex organization of the brain [1]. Such a complexity arises from several, integrated, segregated, and distributed networks around critical and participating cortical epicenters embedded in their physical space. Modern network neuroscience has led to a paradigmatic improvement in understanding the brain-network organization and has challenged the traditional concept that many neurological disorders involve either focal or widespread alterations.

Combining connectomics and network science allows for the investigation of the topological architecture of brain networks, considering the brain areas and their structural and functional connections [2]. The topology of brain networks can be modelled in form of graphs and can be represented as connectivity matrices where each row or column corresponds to different brain units/elements (nodes), and each element of the matrix indicates the value of the structural, functional, or effective pairwise relation between two nodes (edges) [3,4]. 

During the last decade, several magnetic resonance imaging (MRI) and neurophysiological studies have demonstrated that the brain networks present an intrinsic small-world (SW) architecture, functionally segregated (local clustering) and integrated (global efficiency), which is organized into modules with high clustering and short characteristic path length [5,6,7]. This enables information to travel quickly and efficiently even between far brain structures, as well as to prevent the uncontrolled spread of information across the whole network [8]. It is likely that such aspects of complex organization may subserve several critical functions of the human brain, including consciousness. According to the information integration theory, consciousness is the product of functional interaction of multiple brain structures and depends on the brain’s ability to integrate different complex patterns of internal communication [9].

Investigating the traits of consciousness from the connectomic perspective offers insights on how spontaneous integration and segregation of information relate to the human cognition and how such complex organization may be affected in different conditions [10]. 

The complex patterns of neuronal activity associated with consciousness are severely damaged in patients with chronic disorders of consciousness (DoC) [11,12,13,14,15,16,17,18]. This has been put in relation to the brain damage-dependent connectivity breakdown between and within fronto-parietal regions influenced by specific circuit modulations of the thalamus [19]. In terms of this respect, PET activation studies demonstrated that patients with unresponsive wakefulness syndrome (UWS) present a global disconnection syndrome in which higher-order association cortices are functionally disconnected from primary cortical areas (unstable functional communication). By contrast, functional Magnetic Resonance Imaging (fMRI) studies showed that the long-range cortical networks associated with language and visual processing are preserved in patients with a minimally conscious state (MCS) [20,21]. Regarding this aspect, the MCS was recently subcategorized based on the complexity of patients’ behaviors: MCS+ describes high-level behavioral responses (i.e., command following, intelligible verbalizations or non-functional communication) and MCS− describes low-level behavioral responses (i.e., visual pursuit, localization of noxious stimulation, or contingent behavior such as appropriate smiling or crying to emotional stimuli). In addition, it has been also hypothesized that this connectivity breakdown in UWS and MCS is paralleled by a dysfunctional aberrant limbic hyperconnectivity, which may indicate the existence of self-reinforcing loops engaging spatially close areas of residual neuronal activity [22]. Indeed, Di Perri et al. showed a pathological positive between-network hyperconnectivity in DoC patients, as well as a negative default mode network (DMN) connectivity (a form of between-network connectivity) in these patients who emerged from DoC [23]. 

Nevertheless, there is no clear correlation between the degree of connectivity breakdown and of behavioral impairment, with a consequentially high misdiagnosis rate [24,25]. The analysis of complex brain networks using graph theoretical methods on fMRI and electroencephalography (EEG) datasets [26,27,28,29,30,31,32] could provide clinicians with more objective tools for reaching better diagnosis, prognosis, and treatments. Regarding this aspect, the stability and organization of brain networks fails when the main nodes with high centrality (so-called hubs) are impaired, with serious consequences for brain network integrity and functioning [33,34,35]. However, the global and local network topological features related to the level of consciousness are still largely unknown. In this regard, Achard et al. showed an extensive re-adjustment of the hub nodes in comatose patients shortly after brain injury [36]. Crone et al. demonstrated that the nodal topology of patients with chronic DoC differs from the healthy brain, especially in many areas of the fronto-parietal network. Moreover, the local efficiency of the precuneus is reduced when comparing patients with UWS and MCS [19]. More recently, altered and anti-correlated topological measures in the fronto-parietal network (FPN) and DMN have been revealed to exist between UWS and MCS [37]. Finally, Chennu et al. demonstrated that some network measures correlate with the behavioral diagnosis and recovery in patients with DoC, thus corroborating the clinical diagnosis, identifying the patients who may benefit from further assessment, and providing an objective prognostic characterization [30]. 

However, aberrant network topology between MCS and UWS patients has been demonstrated only at the nodal level and investigating specific subnetworks (i.e., FPN and DMN) by using sophisticated devices, e.g., fMRI. On the other hand, despite the prominent alterations in brain connectivity/functions and the identification of disturbances in information integration at the system level, affecting arousal and awareness, surprisingly, there is no clear evidence of aberrant network topology features between MCS and UWS patients [38]. However, identifying alterations in the topological features of the functional networks in altered states of consciousness may be a relevant issue, as it could equally offer significant insight concerning DoC differential diagnosis. In fact, consciousness generation, maintenance, and even recovery are associated with connectivity modulation across fronto-parietal regions on a large scale, modulated by the thalamus, which can be captured by topological network measures. To examine the extent to which network measures can be adopted as quantitative markers for DoC differential diagnosis, we applied them to EEG-derived functional connectomes. In addition, we related the discovered EEG-network alterations to deficits in cognitive functioning and conscious processing. In fact, the results of different topological network properties should be combined in a unique interpretative framework. This holistic view can suggest the reasons behind network dysfunctions that are missed by taking the results of topological network properties one by one. Further, this study is focused on functional connectomic data derived by EEG, which, it is worthy to note, is a device easily applicable in the clinical setting. In contrast, the abovementioned studies employed sophisticated and advanced devices, which are not straightforward for DoC diagnosis. 

To summarize, in the present paper, adopting complex network analysis on resting-state EEG data, we aim to quantify and understand the pathological features of the functional connectomes in DoC. We hypothesize that variations in topological proprieties of these functional connectomes could help in discriminating between patients with UWS and MCS, and could provide useful information on the functional connectivity patterns which characterize this difference from a neuroscientific perspective.

## 2. Materials and Methods

### 2.1. Participants

Forty-five patients with DoC attending the Intensive Rehabilitation Units of the IRCCS Centro Neurolesi Bonino Pulejo, Messina (Italy) between September 2017 and March 2018, were recruited according to the following inclusion criteria: (i) MCS and UWS/vegetative state diagnosis [39]; (ii) no systemic diseases; (iii) no history of psychiatric diseases; (iv) no intake of drugs affecting cortical activity except for L-Dopa, baclofen, pain-killers, and antiepileptic drugs. Therefore, 31 patients were subjected to daily Coma Recovery Scale-Revised (CRS-R) [39] assessment for one month, after which they underwent EEG recording in a resting condition. Six patients had to be excluded due to poor EEG data quality; thus, 25 EEG datasets were analyzed.

The CRS-R is a reliable tool enabling distinguishing patients in UWS from those in an MCS. The CRS-R consists of 29 hierarchically organized items divided into six functional subscales addressing auditory, visual, motor, oro-motor, communication, and arousal processes [39].

The demographic and clinical characteristics for all participants are summarized in Table 1. The entire study was approved by the Institutional Review Board of IRCCS Centro Neurolesi Bonino Pulejo (Messina, Italy), and written informed consent was signed from the legal surrogate of each patient.

### 2.2. EEG Recording and Processing

EEG recordings were carried out with a digital EEG machine from 19 electrodes (Fp1, Fp2, F3, F4, C3, C4, P3, P4, O1, O2, F7, F8, T3, T4, T5, T6, Fz, Cz, Pz, Oz, A2, and A1) positioned in accordance with the International 10–20 system. Two separate channels, vertical and horizontal electrooculograms, were used to monitor eye blinks. Impedance was kept below 5 kΩ, and the sampling rate frequency was set up at 256 Hz. EEG signals were measured at rest, for at least 5 min between 9 a.m. and 11 a.m., with the patient lying in a semi-supine position with the eyes closed, and no task was executed.

Data were analyzed by using Matlab R2015b software (MathWorks), via scripts based on EEGLAB 11.0.5.4b toolbox (http://www.sccn.ucsd.edu/eeglab). Recordings were band-pass-filtered (Finite Impulse Response—FIR, 0.7–30 Hz), and re-referenced to both mastoids. EEGLAB’s plugin CleanLine was employed to identify and remove significant sinusoidal artefacts from the scalp channels using a frequency-domain (multi-taper) regression technique with a Thompson F-statistic. Eventually, bad channels were then rejected following objective and controlled criteria with the EEGLAB plugin clean_rawdata and the continuous data were further corrected using the Artifact Subspace Reconstruction method that removes non-stationary high-variance signals from EEG. The removed channels were then interpolated with the spherical method.

An Independent Component Analysis (ICA) procedure performed in EEGLAB Infomax ICA algorithm was run on the continuous data in order to remove eye movements, muscle contraction, EKG activity, and subgaussian sources of activity. Artefact-free EEG was thus segmented into 2-s epochs.

### 2.3. Cortical Source and Functional Connectivity Estimation

Brain connectivity was computed by eLORETA software [40] on 84 regions-of-interest (ROIs) defined according to the 42 Brodmann areas (BAs: 1, 2, 3, 4, 5, 6, 7, 8, 9, 10, 11, 13, 17, 18, 19, 20, 21, 22, 23, 24, 25, 27, 28, 29, 30, 31, 32, 33, 34, 35, 36, 37, 38, 39, 40, 41, 42, 43, 44, 45, 46, and 47) for the left and right hemispheres. ROIs were needed for the estimation of the electric neuronal activity used to analyze brain functional connectivity. The eLORETA [40] software uses a realistic head model [41] based on the MNI152 template, with the three-dimensional solution space restricted to the cortical gray matter as determined by the probabilistic Talairach atlas [42]. The intracerebral volume was partitioned in 6239 voxels at a 5 mm spatial resolution and the anatomical labels corresponding to BAs were reported using the neuroanatomical Montreal Neurological Institute (MNI) space, converted to the Talairach space [43].

Node-wise synchronizations were estimated through the peak lagged phase synchronization (LPS) extracted by the “single nearest voxel” option across all time and frequency bins within six independent EEG frequency bands [44], that is, δ (2–4 Hz), θ (4–8 Hz), α1 (8–10.5 Hz), α2 (10.5–13 Hz), ß1 (13–20 Hz), and ß2 (20–30 Hz), for each subject. LPS estimates the non-instantaneous information exchange across networks as the phase or the imaginary component of channel-wise cross-spectra, being less susceptible to artefacts and volume conduction [45], which can be written as:(1)$$\varphi_{xy}^{2}\left(\omega \right)=\frac{{\left\{Im\left[f_{xy}\left(\omega \right)\right]\right\}}^{2}}{1-{\left\{Re\left[f_{xy}\left(\omega \right)\right]\right\}}^{2}}$$
where $\varphi_{xy}^{2}\left(\omega \right)$ represents the LPS between signals in the frequency domain; ω is the discrete frequency considered; *x* and *y* are the EEG sources; *Re* and *Im* denote the real and the imaginary parts of a complex function, respectively; and x(ω) and y(ω) indicate the discrete Fourier transforms of the two signals of interest *x* and *y* at frequency ω, respectively. The general lagged phase synchronization is defined as the partial coherence between the normalized complex-valued stochastic variables (x(ω), y(ω)) with the zero-lag effect removed [45]. The LPS values are bounded in the range from 0 (no synchronization) to 1 (perfect synchronization).

The result of this process is a weighted network (for each subject and each frequency band) represented as an 84 × 84 adjacency matrix C = (c_ij_), where each node is represented by a given BA and each edge as the node-wise functional connectivity estimated by the LPS normalized by the maximum value of the matrix.

Proportional thresholding was performed on the functional connectivity matrices by selecting the proportional thresholded (PT%) strongest connections of the derived LPS-weighted connectivity matrix and setting these connections to 1, whereas all other connections were set to 0. Proportional thresholding of a functional connectivity matrix thus resulted in a binary graph with a density of PT%. We examined a range of levels of PT from 35% to 1% in steps of 1% so as to depict the trend of the topological measures in UWS and MCS patients across the whole range of thresholds. Considering the robustness of the curves and results, we have then decided to use the specific cut-off of PT, 20%, to illustrate results, in line with previous works [46,47]. The topological measure curves across the whole range of thresholds explored are provided in Appendix A. Then, we checked whether disconnected nodes were present after PT. If disconnected nodes were present in the functional connectomes after PT, they were systematically removed before the computation of the topological measures. In addition, we computed the overall functional connectivity of each connectome as the mean of the absolute values of the edge weights (strengths) in a connectome. We performed a statistical test to compare (for each of the 6 EEG frequency types) the overall FC in UWS and MCS patients in order to investigate the extent to which a biasing effect for overall FC was present in the data. The result of this analysis is provided in Appendix A. In this way, it is possible to exclude that the altered topology of a network is dependent from the FC between groups.

### 2.4. Topological Network Analysis

In complex network analysis, a topological network measure quantifies, by means of a unique numerical value, the extent to which a certain mechanism of organization (or topological feature) influences the network connectivity. Network measures can be stochastic or deterministic.

Stochastic measures involve random procedures during their computation, for example, the generation of randomized networks (a null model of a given network) based on some topological characteristics preserved from the original network. These randomized networks are used to evaluate the prevalence that a certain topological mechanism of organization shows in the original network in respect of the randomized model of the same network. The stochastic process according to which these null models are created induces the stochasticity in the output of the measure, and a good practice is therefore to perform the computation multiple times and to analyze the average behavior and standard error of the measure over the different repetitions. Three stochastic measures that will be presented are: small-worldness, modularity, and structural consistency.

Deterministic measures, instead, are based on the direct quantification of a considered network topological feature (or rule of organization), e.g., node degree. The randomized networks, which represent the null models, are not required to evaluate deterministic measures; hence, the numerical value associated to the measure computed for a given network is always the same. Six deterministic measures that will be presented are: characteristic path length, average clustering coefficient, global efficiency, local efficiency, node betweenness centrality, and local-community-paradigm correlation. A detailed description of both stochastic and determinist measures is provided in Appendix A, whereas a brief and conceptual introduction of the same measures is offered below.

In addition, a topological measure can be either local or global. It is local if it makes a statistical evaluation of local topological information in the neighborhood of a node or a link. It is global if it makes a statistical evaluation of global topological information that emerges from nodes or links that are not in a neighborhood. Note that for neighborhood we intend the ensemble of nodes that are first-neighbors of a given node or edge. We will specify whether a measure is local or global in detailed descriptions below.

### 2.5. Stochastic Measures


-The small-worldness [48,49] was proposed for the characterization of a given network as SW, meaning that it exhibits a high average clustering coefficient and a low characteristic path length [50]. It relies on comparing a given network with an equivalent random network and a lattice network on the basis of the average clustering coefficient, a local measure, and the characteristic path length, a global measure.-The modularity (Q) [51,52] is a global measure that indicates the possible presence of segregated modules or communities in a network. In networks with high modularity, the modules tend to interact densely within themselves but sparsely or not at all between each other.-The structural consistency [53] is a global measure that quantifies the link predictability of a complex network. The link predictability characterizes the inherent difficulty to predict the missing or non-observed links of a network, regardless of the specific algorithm used for the prediction.


### 2.6. Deterministic Measures


-The characteristic path length (L) [49,50] is a global measure and describes the average of the shortest path lengths between all the pairs of vertices. A small value of L in a connectome means that the information flow between the nodes across the network is facilitated, and that the nodes are able to exchange messages between each other easily. In other words, the nodes across connectomes are functionally convergent.-The average global efficiency (E) [54,55] is a global measure that quantifies how efficiently the information is exchanged within the network. The average local efficiency instead reflects the extent of integration between the immediate neighbors of a given node. In this way, local efficiency can be considered a generalization of the clustering coefficient that explicitly takes into account paths.-The average clustering coefficient (ACC) [50] is a local measure and offers an average evaluation of the cross-interaction density between the first neighbors of each node in the network.-The average node betweenness centrality (ANBC) [56] is a global measure based on the node betweenness centrality, an indicator of node centrality that evaluates how crucial a particular node is in maintaining a path of optimum information flow between any other pair of nodes.-In contrast to the existing node-neighborhood-based local measures, a new strategic shift has been introduced recently in which the focus is no longer only on groups of nodes and their common neighbors, but also on the organization of the links between them [57]. This new idea inspired a theory, which is known as the Local Community Paradigm (LCP) theory, and is valid both in monopartite [57,58] and in undirected unweighted bipartite networks [59,60]. The LCP theory was proposed to mechanistically and deterministically model local-topology-dependent link-growth in complex networks, and states that for modelling link prediction in complex networks, the information content related with the common neighbor nodes (CNs) of a given link should be complemented with the topological information emerging from the interactions between them. The cohort of CNs and their cross-interactions—which are called Local Community Links (LCLs)—form what is called a local community. This first part of the theory inspired the Cannistraci’s variation of the classical CN-based similarity indices for link prediction, named also LCP-based link predictors. For details, refer to [57,58,59,60]. Furthermore, the LCP theory holds that in many complex network topologies, the number of CNs of each link in the network is positively correlated with the respective number of LCLs. This second part of the LCP theory motivated a new network measure called local-community-paradigm correlation (LCP-corr) [57,59,60], which is a local measure that represents an exception with respect to the majority of the previous ones, for two main reasons. Firstly, it is not related with only the node neighborhood but with the node/link neighborhood. Secondly, the general statistic used to obtain a unique value is not the average but the Pearson correlation. The formula for computing the LCP-corr is:
(2)$$LCPcorr=\frac{cov\left(CN,LCL\right)}{\sigma_{CN}\cdot \sigma_{LCL}}$$
with *CN* > 0, where *cov* indicates the covariance operator and σ is the standard deviation. In normal conditions, brain connectomes follow LCP organization [57], and therefore they are characterized by high LCP-corr (usually > 0.8). A recent study [61] investigated the LCP organization in time-varying brain functional connectomes of a rat model of persistent peripheral neuropathic pain, obtained by means of local field potential and spike train analysis. LCP-corr was employed to quantitatively investigate the rewiring mechanisms of the brain regions responsible for development and upkeep of pain along time, from three hours to 16 days after nerve injury. The time trend (across the days) of LCP-corr was correlated with a behavioral test for rat pain, and surprisingly this analysis showed very high statistical correlations (higher than 0.9, the maximum value being 1) of LCP-corr with the behavioral test [61].


### 2.7. "Functional Network Topology" Exploration: An Overview

In this section, we will offer an example of how to interpret complex networks analysis in functional brain connectomes, and how this might differ from structural connectomes. For instance, functional segregation can relate to the local-derived concept of average clustering coefficient of the functional connectome. Despite such spatial segregation, the brain demonstrates also global functional integration in various aspects, for instance, combining specialized information from different regions to provide a unitary behavioral output, which reflects a coherent response to the integration and combination of multiple local processes. Functional integration can relate to the global-derived concept of average path length in the functional connectome. Specifically, in the structural connectomes, path length means combination of nodes and links resulting in physical information flow; whereas in the functional connectomes, path length means a sequentially coherent statistical relation between subsequent regions, and might not be always supported by physical information flow through anatomical connections.

### 2.8. Network-Based Statistic of Network Connectivity

In order to identify eventual subnetworks in which the functional connectivity is altered in the UWS or MCS patients, we used the Network-Based Statistics (NBS), a powerful approach allowing for multiple hypothesis tests at the level of interconnected subnetworks, controlling the family-wise error when performing analyses associated with a particular effect or contrast of interest [62]. NBS, in fact, overcomes some of the limitations of the generic procedure (such as the false discovery rate) which computes statistical tests and corresponding *p*-value independently for each link and considers exclusively the strength of that link. Briefly, NBS performs a mass univariate testing in order to identify the connections exceeding a test statistic threshold belonging to a given connected component. Finally, a corrected *p*-value is computed for each component using the null distribution of maximal connected component size, which is empirically derived via a nonparametric permutation method. For a complete description of the procedure, the reader can refer to [62].

### 2.9. Statistical Analysis of Data

A Mann–Whitney U (MW) non-parametric test was used to compare MCS and UWS in each topological network measure. The significance level was set at *p* < 0.05. Correction for multiple hypothesis testing has not been performed when comparing whole-brain topological network measures, since we are not doing a feature selection, but we are instead looking at the performance of different measures in separating the two groups. In this case, we also computed the area under the ROC curve (AUC) and the area under the precision-recall curve (AUPR). On the other side, when testing the single nodes that are significantly different, we performed a feature selection and therefore we opted for a Benjamini-Hochberg correction over the brain areas investigated to control the false discovery rate in multiple hypothesis testing. To determine the significance levels of altered connectivity networks in NBS analysis, we first performed a two-sample *t*-test at each edge independently to test for significant differences in the value of connectivity between MCS and UWS. A primary threshold (*p* = 0.05, *t* = 2.6, two tailed *t*-test) was applied to form a set of suprathreshold edges among which any connected components and their size (number of edges) could then be determined. Next, the statistical significance of the size of each observed component was then evaluated with respect to an empirical null distribution of maximal component sizes obtained under the null hypothesis of random group membership (20,000 permutations). The subnetworks were considered statistically significant at a corrected level of *p* < 0.05. Finally, we computed both linear (Pearson) and nonlinear (Spearman) correlations of each topological measure (both at the whole-brain and single-node levels) versus the behavioral measure (i.e., CRS-R score). This analysis explained the extent to which a certain topological measure is able to capture topological features of which variations in the functional brain networks are correlated with the presence of consciousness. The same procedure was applied to post-hoc selected single edges in the subnetworks identified by the NBS analysis versus the CRS-R score. In brief, these steps spot markers for the rewiring correlates of consciousness in the functional connectomes of DoC patients. The correlations were considered statistically significant if the related *p*-values were less than 0.05.

## 3. Results

Table 1 reports the demographic and clinical data of the patients with MCS and UWS included in this study. No significant differences were found between the two groups, with the exception of the CRS-R. Appendix A shows that no difference in the overall functional connectivity strength was found between the UWS and MCS patients.

### 3.1. NBS Analysis

The NBS identified one subnetwork of decreased functional connectivity in UWS patients compared to in the MCS patients in the ß1 frequency (*p* = 0.004, corrected for multiple comparisons) (Figure 1, and Appendix A). The subnetwork consisted of fifty-four edges connecting thirty-two different cortical areas. Interestingly, apart from a few intra-hemispheric pathways linking limbic regions with frontal and parietal areas, these patterns of reduced connectivity mainly involved the interhemispheric fronto-parietal network (Figure 1).

The single network edges that were identified in the NBS analysis were post-hoc selected for further clinical–electrophysiological correlations. Significant correlations between the strength of the following connectivity patterns and the clinical diagnosis were found: left orbital part of the inferior frontal gyrus–right visuomotor area (Pearson’s Rho = 0.53, *p* = 0.007; Spearman’s Rho = 0.54, *p* = 0.006), left pregenual area–right primary somatosensory area (Pearson’s Rho = 0.51, *p* = 0.009; Spearman’s Rho = 0.47, *p* = 0.01), right pregenual area–right pars opercularis (Pearson’s Rho = 0.49, *p* = 0.01; Spearman’s Rho = 0.49, *p* = 0.01), left dorsolateral and medial prefrontal cortex-right subcentral area (Pearson’s Rho = 0.51, *p* = 0.01; Spearman’s Rho = 0.44, *p* = 0.02), left pregenual area–right supramarginal gyrus (Pearson’s Rho = 0.48, *p* = 0.01; Spearman’s Rho = 0.50, *p* = 0.01). Appendix A reports the clinical–electrophysiological correlation coefficients for every single edge belonging to the dysconnectivity subnetwork identified in UWS by the NBS analysis.

### 3.2. Whole-Brain Network Topology Measures

In the network topology, the two groups showed significant differences in some measures only in the ß1 band (Table 2).

Among the deterministic measures, post-hoc tests indicated increased values of LCP-corr and clustering coefficient in the brain networks of the patients with UWS (*p* = 0.03). Among the stochastic measures, both the UWS and MCS patients showed positive values of SW indices, close to zero. In particular, there was a significant group difference for both *SW**ω* (*p* = 0.01) and *SW**ω-E* (*p* = 0.02), with the UWS patients having indices closer to zero. Additionally, the local efficiency was significantly different between the two groups in favor of UWS patients (*p* = 0.01). The other measures did not differ at the threshold level. However, there were no significant clinical–topological correlations at a within-group level, as indicated by the poor and non-significant correlation between CRS-R scores and the topological network measures (Table 2). Since we looked at the performance of whole-brain topological network measures in separating UWS from MCS patients, we have also computed the AUC and AUPR. Table 2 shows that these two evaluation measures confirm the results previously discussed for the MW p-value, providing high levels of separation between the different states of consciousness, with the best performances given by the *SW**ω* (AUC = 0.81; AUPR = 0.73) and by the local efficiency (AUC = 0.80; AUPR = 0.72).

### 3.3. Nodal Measures

Compared with the MCS, the UWS patients showed an increased nodal degree in several limbic and temporo-parieto-occipital regions. The ß1 frequency showed the most widespread alterations with lower degree in many frontal regions in addition to enhanced degree in the posterior cingulate cortex. Figure 2 shows the several degree changes found in the connectomes of the UWS and MCS patients.

Similar alterations were found for the nodal betweenness centrality. Following the same pattern of the nodal degree, many alterations for the betweenness centrality were found in the ß1 band. In particular, higher betweenness centrality was found in the visual-related and posterior cingulate area, as well as lower betweenness centrality in many bilateral frontal regions (Figure 3).

Also for the clustering coefficient, the ß1 band showed the most widespread alterations across many frontal, parietal, and cingulate regions with a higher clustering coefficient in UWS compared with that in the MCS patients (Figure 4). 

After performing multiple hypothesis testing correction using the Benjamini-Hochberg procedure, no significant results have been found for single node topological network measures between UWS and MCS patients (Appendix A). Finally, statistically significant correlations, despite low, between single-node topological measures and the CRS-R score have been found (Appendix A).

## 4. Discussion

To the best of our knowledge, this is the first study investigating brain network topology in patients with DoC using network topology measures on small EEG datasets. There are four main findings: (i) patients with UWS and MCS differ concerning the interhemispheric fronto-parietal connectivity in the ß1 frequency range; (ii) the network topology properties of EEG brain networks enable differentiating patients with UWS from those with MCS at a group level; (iii) regions of fronto-parietal networks differ in their network properties with a consequential impairment of local information transfer; and (iv) behavioral responsiveness (as per CRS-R) is significantly, despite poorly, correlated with some topological measures.

The NBS analysis demonstrated the existence of disconnected subnetworks mainly involving the long-range, fronto-parietal, interhemispheric connections in the patients with UWS as compared to those with MCS, as a possible marker of awareness. This issue agrees with previous works that identified decreased interhemispheric functional connectivity in subjects with impaired awareness [63,64,65]. Moreover, our data further confirm that a putative neural basis of (un)awareness relies on the involvement and activation of the fronto-parietal network driving widespread functional connectivity changes across the brain, even those related to multisensory integration, top-down processing, and awareness. Recent works highlighted the role of beta/gamma frequency range in the posterior regions as a neural correlate of conscious contents [66,67]. We extended these findings by highlighting the fronto-parietal, interhemispheric connectivity in the ß1 band as a potential correlate of the level of behavioral responsiveness estimated through the CRS-R.

Noteworthy, NBS analysis did not discriminate patients with DoC at an individual level. Indeed, two of the patients with UWS showed a topological measure profile matching that of patients with MCS. This is not surprising, as the NBS approach offers high sensitivity in detecting disconnections in a network by exploiting the extent to which the abnormal connections are interconnected [68]. However, NBS data are not specific to any network topological measure (i.e., they cannot offer information related with a particular property of the topology that differs between the groups) [62], although the identified subnetworks showed significant between-groups differences.

Therefore, the deeper understanding of brain topology and the extent to which a network holds certain topological characteristics (e.g., integration and differentiation) are important to identify the key elements supporting awareness, in particular at an individual level [69,70,71]. Regarding this respect, complex network topology analysis of the functional connectomes allows investigating both the global and local topological organizations, as well as specific connections between the regions. Thereafter, the abnormalities in functional brain network topology and connectivity can be correlated with clinical scores. Thus, we may use the network features to distinguish patients with MCS from those with UWS.

Resting-state connectivity and local network topology (e.g., “Extrinsic”/”task-positive” and “default-mode” networks (DMNs)) have been well described [19,29,30,70,72,73,74], with particular regard to the DMN, a subset of regions that are deactivated during externally oriented tasks and that are negatively correlated with the degree of behavioral impairment [75,76]. On the other hand, whether the patients with DoC differ concerning the deterioration of topological organizations in the functional networks has not yet been completely described with small EEG datasets [19,29,30]. Some studies identified significant changes in topological properties between healthy individuals and patients with DoC, but not between patients with UWS and MCS [19,36]. Hence, how different brain areas are integrated and segregated for communication and specialized processing remains partially unknown at both between- and within-group levels.

In order to identify the global and local network properties that could help to differentiate patients at between- and within-group levels, we first had to demonstrate that both DoC groups exhibited a SW topology. Both DoC groups had highly dense connections between nearest neighbors and a low average path length, thus confirming that complex brain networks have SW properties [77,78,79,80]. Contrary to the available data, we found a significant difference between the two groups. This discrepancy may depend on both methodological (EEG vs. fMRI) and sample selection patients (including age, disease duration, and etiology).

At a network topology level, we found that the patients with UWS showed higher values of LCP-corr (that estimates the size of local communities and their information, which can be used as an indicator of self-organization in complex networks), clustering coefficient (a measure of the degree to which nodes in a graph are forming a cluster with their neighbors), and local efficiency (which measures the information exchange limited to direct neighboring nodes, with respect to a node if interested) than those with MCS. Altogether, these data indicate a trend to form small isolated and disconnected networks in the former group. In other words, there is an imbalance between segregation and integration (which is critical to maintain the high level of functioning of human brain networks) [35] greater in patients with UWS than in those with MCS. These data are also in keeping with former studies using different approaches, which outlined a large-scale connectivity breakdown as the main responsible for awareness impairment, with particular regard to the alpha frequency range when using EEG [30,81,82,83,84]. We extend these findings by highlighting the role of specific topological measures within the ß1 frequency range as potential correlates of the level of behavioral responsiveness estimated through the CRS-R.

This is the first time that LCP-corr is measured in patients with DoC. We opted to use such a measure because it indicates when the network architecture is facilitating both the rapid delivery of information across the various network modules and the local processing (i.e., high LCP-corr values). Contrariwise, non-LCP networks (i.e., LCP-corr values < 0.4) characterize stunned and energetically expensive connections. Non-LCP networks have indeed the characteristic of weak interactions between the nodes. In normal conditions, brain connectomes follow an LCP organization [57]. High LCP-corr values suggest that a lot of local-community links are taking place, thus resulting in a more dynamic self-reorganization, i.e., new links are added between common neighbors. In keeping with the ability of LCP-corr to capture this kind of local information in the network topology, to investigate the connectivity between the neighbors of a link could be an appropriate strategy to identify the local network remodeling associated with awareness generation and maintenance. Our results suggest that in altered states of consciousness, the local community organization of the network is preserved (LCP-corr > 0.8). However, the network tends towards a more community-oriented organization in patients with UWS (higher values of LCP-corr), pointing out a gradual and cumulative enrichment of neural connections inside the same local community.

It is worth noting that the LCP theory derives from a purely topology-inspired interpretation of the Hebbian learning rule: neurons that fire together wire together [85]. The Hebbian theory assumes that different engrams (memory traces) are consolidated by neuronal populations that are co-activated within a given network. Therefore, it is reasonable to ask how to interpret the concept of wiring together. The first interpretation is the connectivity reinforcement between neuronal cohorts that fire together, while the second interpretation is the rise of new connections between non-interacting neurons already embedded in an interacting population. Several studies demonstrated that some kinds of learning involve synaptic changes with an unaltered number of neurons [86,87], thus proving the first interpretation of the Hebbian learning theory. The second interpretation, instead, has been recently formalized as a purely topology-inspired problem of topological link prediction in complex networks [57]. Briefly, the local-community topological organization plays a major role in creating a physical and structural “energy barrier” that allows neuronal populations to preferentially fire together within a given community adding new links inside that community. Based on these simple premises, it can be reasonable to assume that the higher tendency towards a local-community-oriented topological organization in UWS may represent an epiphenomenon of diffusely emergent (possibly) dysfunctional connections, resulting in aberrant self-reinforcing loops. Therefore, such an epi-topological phenomenon may play a critical role in the interactions with the external environment, learning and information storage in long-term memory, and finally awareness.

A second important finding is that the clustering coefficient displayed higher values in patients with UWS than in those with MCS, thus representing another marker of behavioral responsiveness and potentially awareness. The clustering coefficient provides an average evaluation of the cross-interaction density between the first neighbors of each node in the network [50], thus being a measure of network segregation. Although at a first glance, such a behavior can seem an unexpected result, the increased neighborhood connectivity can represent a possible compensatory mechanism that is triggered by the deterioration of the long-range cortical-thalamo-cortical connectivity, and therefore of the top-down modulation mechanisms from higher-order cortical areas to sensory-motor integration networks in patients with UWS [88,89]. Interestingly, a negative correlation between clustering coefficient and the robustness of a network has been demonstrated [90]. This means that increasing the clustering of a network results in a reduced proportion of the edges between topologically far network’s nodes. Thus, the network could be easily broken apart into different components. In other words, higher clustering coefficient values make the networks of patients with UWS more vulnerable to random failure in brain connectivity.

Last, higher local efficiency values were found in the patients with UWS than with MCS. It is worthy to remember that awareness levels depend on the global functional connectivity with a reduced network segregation, which facilitates the information transfer between topologically remote modules [91]. The average local efficiency quantifies the ability of fault tolerance of the network measuring the information exchange of the subnetwork consisting of itself and its all direct neighbors [77]. Therefore, the higher values of average local efficiency found in UWS compared with in MCS patients suggest that the functional brain networks of patients with UWS are topologically organized in a way that maximizes the segregation of neural processing.

We may speculate that such an aberrant network topology reflects the engagement of residual neural activity in short-range self-reinforcing loops that, in turn, may lead to a disruption of the connectivity patterns important for awareness and processing of multimodal information [20,21,92]. We also may hypothesize that this topological rewiring is a complementary feature of (un)awareness related to the involvement of the fronto-parietal networks.

The available data in the literature left partially unsolved the problem of whether a correlation exists between CRS-R scores and network topology measures. Even though topological measures provided a between-group differentiation, CRS-R scores and these measures were poorly correlated [19]. This may depend on the fact that massive brain damage, as in patients with DoC, may lead to a comparable impairment of network measures in keeping with the reduction of connection density and the reorganization phenomena throughout the brain [19,36]. We thus hypothesized that the topological organization required for consciousness at a level as low as that existing in patients with DoC could be reflected only when assessing simultaneously different topological network measures (LCP-corr, clustering coefficient, local efficiency, etc.). This is a very important point concerning DoC diagnosis and consequently management. In fact, the clinical presentations of patients with MCS and UWS can be relatively similar in case of borderline CRS-R scoring (i.e., 6-to-9), although having different levels of awareness, and discriminating between reflexive and willful behavior can be difficult [93]. There are, in fact, many biasing sources [25,94]. Such clinical conditions have been labeled as non-behavioral MCS (MCS*), cognitive-motor dissociation, Functional Locked-In Syndrome, Vegetative State with hidden consciousness or with preserved islands of consciousness, in which a behaviorally unresponsive patient is covertly aware, i.e., aware but unable to manifest it (owing to, e.g., a severe motor impairment, with particular regard to the motor cortico-thalamo-cortical circuits) [88,89,95,96,97,98,99].

Once identified the significant group difference in terms of topological changes, we sought whether the brain networks of the two groups differed also at the single nodes level. Thus, we computed the nodal degree, the nodal clustering coefficient, and the nodal betweenness centrality for each node of the network in the ß1 band (that was the unique frequency range in which we found network topology between group differences). The two groups significantly differed concerning several regions of the frontal, parietal, and cingulate regions in the abovementioned topological characteristics (see Figure 2, Figure 3 and Figure 4). The fact that the correction for multiple hypothesis testing nullifies any evidence of significant *p*-values (Appendix A) does not mean that no differences exist between UWS and MCS patients at the nodal level, indeed this only suggests that the results obtained without correction should be considered with a grain of salt and caution. Such results offer an indication that needs further investigation, but at the moment they cannot be considered as markers to discriminate the different states of consciousness.

### Limitations

Owing to the intrinsic limits of the methodology employed in our study, we can only speculate on the role played by the thalamus on our data. Given that our findings suggest that the patients with UWS exhibit an aberrant network organization both at the whole-brain network topology and at the nodal level as compared to the MCS patients, the involvement of the thalamus in sustaining such network aberrations is straightforward. Indeed, the thalamus is an integrative relay with different frontoparietal networks, thus representing an important node of the network involved in generating conscious awareness [100,101].

Surprisingly, our findings referred solely to ß1 frequency range. Even though there were some minor changes in every band, including alpha and delta [30], these did not reach the significance threshold and were not of unique interpretation. This fact may depend on our methodological approach (including thresholding, electrode displacement, and ROIs selection). Moreover, there is growing knowledge of the important roles of ß1 frequency range in selective attention and large-scale neuronal integration [102,103]. Indeed, beta oscillations are described to be associated with a steady state of the motor system, i.e., they signal the “motoric status quo” [102,104]. Regarding this respect, ß1 frequency range is tonically pushed up to respond to environmental stimuli even though in an almost unintentional way in patients with UWS, and somehow purposefully in patients with MCS [105,106]. Moreover, beta-band oscillations (13–20 Hz) are mainly located in the parietal regions [99], where we found the most relevant local connectivity changes influencing brain topology. Further, the cerebral cortex of patients with DoC is tonically active owing to the deafferentation following thalamo-cortical degeneration [107,108,109], which impairs GABAergic tone, of which entity correlates with beta oscillation magnitude [110,111]. Last, beta oscillatory parameters have been shown to reveal changes in the excitatory–inhibitory balance in M1, which could be used as a marker of plasticity in the brain [112].

## 5. Conclusions

Discriminating between different states of consciousness remains a challenging matter, also considering the wide spectrum of clinical features characterizing patients with altered levels of consciousness. We propose that bedside clinical assessments paired with resting EEG, also when only small EEG datasets are available, could improve our ability to differentiate at group-level patients with DoC [30,113]. In conclusion, whole-brain network topology measures represent at the moment the only computational method that offers a marker to significantly distinguish UWS and MCS. Concerning the complexity of the network analysis, it is worth noting that its steps are largely automated from an application perspective, also somehow including inspecting and identifying noisy data and independent components.

Moreover, our findings confirm and further extend the available data on the connectivity breakdown as a main responsible unawareness in the patients with DoC. Last, our approach further promotes the clinical utility of the resting paradigm for group and single-patient diagnostics and potentially for evaluating the effectiveness of specific interventions, including non-invasive neuromodulation.

## Figures and Tables

**Figure 1 jcm-08-00306-f001:**
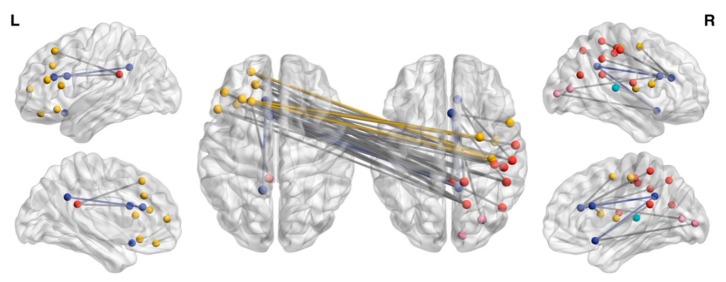
Interhemispheric frontal-parietal subnetwork disconnectivity in UWS. The figure shows the subnetwork with decreased connectivity in UWS compared to in MCS patients in the ß1 (*p* = 0.004, corrected for multiple comparisons), identified by the Network Based Statistic (NBS) analysis. The nodes and the links are overlaid to a surface rendering of the brain in two different projections (sagittal, on the left and right sides; axial, the double brain in the center). Yellow nodes indicate the brain regions belonging to the frontal lobe, the red nodes indicate the brain regions belonging to the parietal lobe, the purple nodes indicate the brain regions belonging to the limbic system, and the cyan nodes indicate the brain regions belonging to the occipital lobe. The subnetwork consisted of fifty-four edges connecting thirty-two different cortical areas. Apart from a few intra-hemispheric pathways linking limbic regions with frontal and parietal areas, these patterns of reduced connectivity mainly involved an interhemispheric fronto-parietal network. The yellow edges represent the interhemispheric connections linking nodes belonging to the frontal lobe, and the purple edges represent the connections between nodes of the limbic system, whereas grey edges represent the connectivity patterns between nodes belonging to different brain lobes (i.e., fronto-parietal). The brain surface with nodes and edges representation was generated with the BrainNet Viewer.

**Figure 2 jcm-08-00306-f002:**
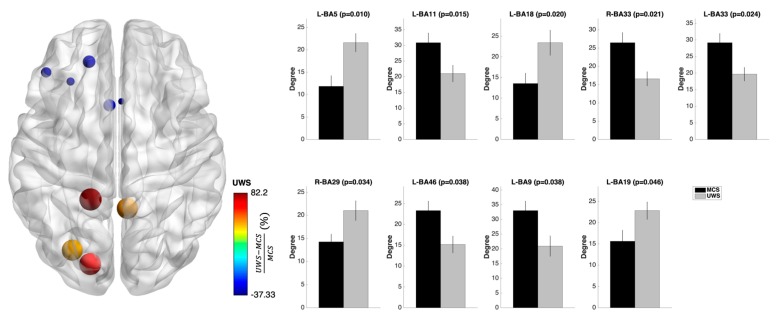
Nodal degree changes at the functional brain networks level of UWS and MCS patients. Compared with the MCS, the UWS patients showed an increased nodal degree in parieto-occipital regions in the ß1 frequencies. In particular, UWS patients showed lower degree in many frontal regions in addition to enhanced degree in the posterior cingulate cortex. On the left side, the nodal degree changes plotted over a glass brain: the size and color of the nodes express the difference in nodal degree (%) between UWS and MCS patients computed as (UWS − MCS)/MCS. On the right side, the mean nodal degrees for UWS and MCS patients are plotted in form of bar plots for each significant brain region. Error bars indicate the standard error of the mean.

**Figure 3 jcm-08-00306-f003:**
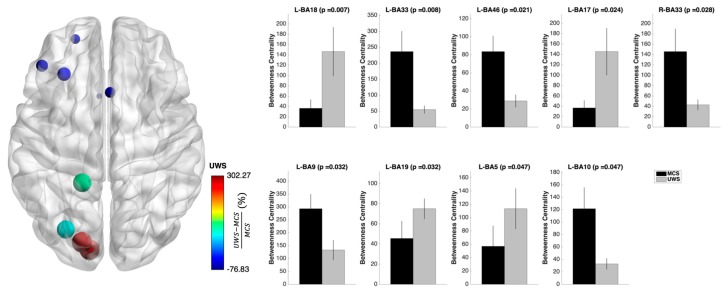
Nodal betweenness centrality changes at the functional brain networks level of UWS and MCS patients. Following the same pattern of the nodal degree, many alterations for the betwenness centrality were found in the ß1 band. In particular, UWS patients showed higher values in the visual-related and posterior cingulate area as well as lower betweenness centrality in many frontal regions. On the left side, the nodal betweenness centrality changes plotted over a glass brain: the size and color of the nodes express the difference in nodal betweenness centrality (%) between UWS and MCS patients computed as (UWS − MCS)/MCS. On the right side, the mean nodal betweenness centrality values for UWS and MCS patients are plotted in form of bar plots for each significant brain region. Error bars indicate the standard error of the mean.

**Figure 4 jcm-08-00306-f004:**
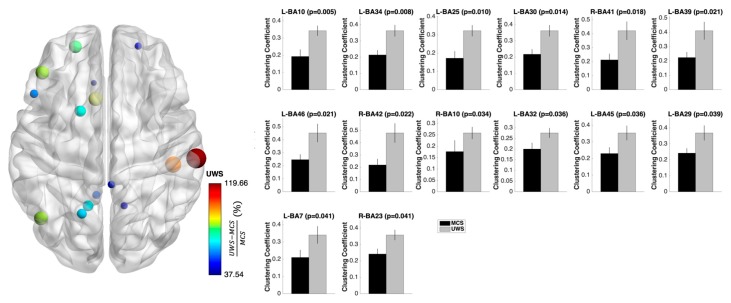
Nodal clustering coefficient changes at the functional brain networks level of UWS and MCS patients. The most widespread alterations for the clustering coefficient were found across many frontal, parietal, and cingulate regions showing higher values in UWS compared with in the MCS patients, suggesting aberrant cross-interactions between the first neighbors of each node. On the left side, the nodal betweenness centrality changes plotted over a glass brain: the size and color of the nodes express the difference in nodal clustering coefficient (%) between UWS and MCS patients computed as $\frac{\mathrm{UWS}-\mathrm{MCS}}{\mathrm{MCS}}$. On the right side, the mean nodal clustering coefficients for UWS and MCS patients are plotted in form of bar plots for each significant brain region. Error bars indicate the standard error of the mean.

**Table 1 jcm-08-00306-t001:** Clinical-demographic characteristics.

DoC	Etiology	Gender	Age	BI Onset	MRI	CRS-R
MCS (*n* = 13)	T	F	57	5	PO_h	12 ± 2
	A	F	54	9	WMH	10 ± 3
	T	M	38	15	FP_h	12 ± 2
	V	M	60	14	TP_IS	11 ± 2
	A	M	36	15	WMH	8 ± 2
	V	F	46	16	BG_h	9 ± 1
	T	M	60	5	F_h	17 ± 3
	T	F	41	8	SAH	12 ± 4
	V	M	57	17	P_IS	9 ± 4
	T	F	42	8	FP_h	16 ± 2
	V	M	65	13	FP_IS	20 ± 4
	A	M	35	7	WMH	18 ± 1
	V	F	54	8	SAH	17 ± 3
	5T 3A 5V	6F 7M	50 ± 10	11 ± 4		13 ± 4
UWS (*n* = 12)	A	F	57	6	WMH	3 ± 2
	T	M	58	16	DAI	4 ± 2
	V	F	62	11	FTP_IS	6 ± 2
	A	F	51	13	WMH	6 ± 2
	T	M	62	6	DAI	3 ± 2
	A	F	61	8	WMH	4 ± 2
	V	M	65	5	FTP_IS	6 ± 1
	A	M	64	18	WMH	7 ± 1
	T	F	56	5	Fb_h	6 ± 1
	A	M	40	12	WMH	5 ± 1
	T	M	41	17	multiple_h	5 ± 2
	T	F	53	7	multiple_h	5 ± 2
	5T 5A 2V	6F 6M	56 ± 8	10 ± 5		5 ± 1
Sample (*n* = 25)	10T 8A 7V	12F 13M	53 ± 12	11 ± 4		9 ± 5
Between-group *p-*value	0.1	0.4	0.1	0.1	0.1	<0.001

DoC: disorders of consciousness; MCS: minimally conscious state; UWS: unresponsive wakefulness syndrome; BI onset: Brain Injury onset; MRI: Magnetic Resonance Imaging; CRS-R: Coma Recovery Scale-Revised; PO: parieto-occipital; _h: haematoma; WMH: white matter hyperintensity; FP: fronto-parietal; TP: temporo-parietal; BG: basal ganglia; F: frontal; SAH: subarachnoid hemorrhage; _IS: ischemia; P: parietal; DAI: diffuse axonal injury; FTP: fronto-temporo-parietal; Fb: frontobasal.

**Table 2 jcm-08-00306-t002:** Main effects of group in the network measures and correlations between whole-brain topological measures and Coma Recovery Scale-Revised score in the β1 band.

Measure	UWS	MCS	MW *p*-Value	AUC	AUPR	Pearson Rho	Pearson *p*-Value	Spearman Rho	Spearman *p*-Value
LCP-corr	0.91 ± 0.01	0.84 ± 0.03	0.03	0.75	0.66	−0.21	0.31	−0.30	0.14
E_loc_	0.56 ± 0.02	0.48 ± 0.02	0.01	0.80	0.72	−0.32	0.11	−0.37	0.07
ACC	0.33 ± 0.02	0.27 ± 0.02	0.03	0.76	0.71	−0.32	0.12	−0.32	0.12
SWω	0.48 ± 0.08	0.58 ± 0.09	0.01	0.81	0.73	0.36	0.08	0.38	0.06
SWω-E	0.33 ± 0.08	0.42 ± 0.10	0.02	0.78	0.69	0.32	0.12	0.32	0.12

Topological network measures values for UWS and MCS are reported as mean ± standard error. The Mann–Whitney (MW) *p*-values indicating statistically significant differences between the two groups, as well as the the area under the ROC curve (AUC) and the area under the precision-recall curve (AUPR) are also reported. The table reports both the Pearson’s and Spearman’s Rho and related *p*-values for electrophysiological–topological correlations between whole-brain topological measures and Coma Recovery Scale-Revised. No statistically significant correlations have been found. LCP-corr: local-community-paradigm correlation; ACC: average clustering coefficient; E_loc_: local efficiency; SWω: small-worldness omega; SWω: small-worldness omega efficiency.

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
