# Peer review of "Functional Brain Network Topology Discriminates between Patients with Minimally Conscious State and Unresponsive Wakefulness Syndrome"

_jcm, 2019, doi:10.3390/jcm8030306_

Reviewer 1 Report

I thank the authors for addressing my initial concerns. I feel the manuscript is substantially improved as a result.

I still have one concern regarding multiple hypothesis testing and family-wise error correction. In figures 3, 4, and 5, the authors show the comparison of nodal measures between MCS and UWS across multiple brain areas. P-values are reported in the text and in the figures (above each barplot), but it is unclear if these p-values are before or after adjustment for multiple comparisons. I feel that some form of correction is warranted to ensure that nodes exhibiting different nodal topology do not suffer from multiple statistical tests (across different nodes and across different nodal measures).

Author Response

We thank the Reviewer for appreciating our efforts in improving the manuscript and for the helpful comments.

COMMENT: I still have one concern regarding multiple hypothesis testing and family-wise error correction. In figures 3, 4, and 5, the authors show the comparison of nodal measures between MCS and UWS across multiple brain areas. P-values are reported in the text and in the figures (above each barplot), but it is unclear if these p-values are before or after adjustment for multiple comparisons. I feel that some form of correction is warranted to ensure that nodes exhibiting different nodal topology do not suffer from multiple statistical tests (across different nodes and across different nodal measures).

REPLY: we thank the reviewer for arising this important issue. We have now computed multiple hypothesis testing correction for the comparison of nodal measures between MCS and UWS across multiple brain areas. We have therefore added this in the statistical analysis section, in results and discussion sections.

As we clarified in the draft:

<< correction for multiple hypothesis testing="" has="" not="" been="" performed="" when="" comparing="" whole-brain="" topological="" network="" since="" we="" are="" doing="" a="" feature="" but="" instead="" looking="" at="" the="" performance="" of="" different="" measures="" in="" separating="" two="" groups.="" this="" also="" computed="" area="" under="" roc="" curve="" and="" precision-recall="" .="" on="" other="" single="" nodes="" that="" significantly="" selection="" therefore="" opted="" benjamini-hochberg="" over="" brain="" areas="" investigated="" to="" control="" false="" discovery="" rate="" testing.="">>

The AUC and AUPR values are reported in Table 2.

Kindest regards,

The authors.

Reviewer 2 Report

The authors have addressed all my concerns. 

Author Response

We thank the Reviewer for appreciating our efforts in improving the manuscript.

Kindest regards,

The authors

This manuscript is a resubmission of an earlier submission. The following is a list of the peer review reports and author responses from that submission.

Round  1

Reviewer 1 Report

Functional Brain Network Topology Discriminates between Patients with Minimally Conscious State and Unresponsive Wakefulness Syndrome

Cacciola et al.

Reviewer Summary:

In this manuscript, Cacciola et al. apply graph theory metrics to EEG-based functional connectivity and investigate differences in network topology in a cohort of patients with two distinct Disorders of Conciousness (DoC) -- unresponsive wakefulness syndrome (UWS) and minimally conscious state (MCS). The study identifies a variety of network features that distinguish functional networks of UWS patients from functional networks of MCS patients, and finds a combination of features that explain individual differences in behavioral responsiveness scores.

I am enthusiastic about the overall approach taken, but I do feel that greater care could be taken to motivate and contextualize the findings.

Introduction

1.     It is unclear what prior evidence substantiates the claim that long-range cortical networks associated with language and visual processing imply the existence of unstable functional communication (lines 65-66). Long-range functional and structural connectivity have been long studied for their contribution towards healthy human cognition.

2.     The authors should make clear how the current work is different novel from prior work. As stated between lines 88-90, “aberrant network topology between MCS and UWS patients has been demonstrated…”. How does this study build upon prior network connectivity work that was introduced in lines 80-87?

3.     The behavioral assessment CRS-R is only described in abbreviated form and never defined. A brief description of this assessment (and its full-form name) in the introduction would help the reader.

4.     The overarching hypothesis presented in lines 93-96 could be better motivated and justified. Why do the authors think whole-brain, global network topology could discriminate the two patient groups? Which direction do they anticipate this relationship based on what is already known about DoC?

Methods

5.     In several places throughout the manuscript, the characters for beta and gamma have not rendered correctly (The first instance is on lines 144/145).

6.     An equation for peak lagged-phase synchronization would help orient the reader to the specific connectivity measure that is being used. What range of values does this statistic yield? Is it bounded?

7.     Are the results presented in this manuscript robust to different proportional thresholds (only a PT=15% was shown here). Since a range of PT were explored, I worry about the implications of multiple hypothesis tests conducted across different PT values. A summary figure demonstrating that PT=15 is in some sense “optimal” would be helpful.

Results

8.     The authors find an interesting link between a combination of LCP-corr and ACC and CRS-R score. The main text describes that the relationship was found by pooling LCP-corr and ACC measurements together, but the specific approach was not provided. How are these measures “pooled” or combined to form a single measurement?

9.     The authors motivate the point that the two outlier patients with UWS exhibit a combined LCP-corr/ACC score similar to the MCS category, which supports the feature has clinical potential for identifying UWS patients with covert awareness (Figure 2 caption). It’s unclear how such a statement can be supported with the small sample size and the data shown here. In this scenario, if a conclusion were made about those two patients on the basis of LCP_corr and ACC, would they not be miscategorized as MCS patients?

10.  The subplot titles in figures 3/4/5 are difficult to read. The colorbar corresponding to the node colors should be labeled so that it’s easy to discern that red corresponds to greater values in UWS patients and blue corresponds to greater values in MCS patients.  

Reviewer 2 Report

In this manuscript, Cacciola and colleagues investigated the functional connectivity of unresponsive wakefulness syndrome (UWS) and minimally conscious state (MCS) patients, by using graph-theoretical tools to analyze topological differences between resting state EEG brain networks.

The manuscript is generally clearly written and logically implemented. However, some of the characteristics of this novel method could be more clearly explained and motivated, as detailed below:

- The authors performed proportional thresholding to binarize the EEG functional connectomes, through a broad range [1:1:35]. They refer to PT=15% in the results: what about the broad exploration?

- Related to PT, are they observing a specific cutoff where they can appreciate better the MCS/UWS difference?

- Have they checked if there are disconnected nodes in the functional connectomes after this thresholding procedure? 

- A more conceptual question is related to the use of "path-based" network measurements in a functional connectome graphs. In the case of this work, one edge in a functional connectome (binarized after thresholding) represents high synchronization between two regions (i.e., not a physical or structural connection, but rather a statistical dependency). What does it mean to evaluate the characteristic path-length (or similar measures) of this object? I would like the authors to elaborate more on the meaning of this "functional topology" exploration, especially for EEG brain graphs.

- Can the authors report what is the actual correlation value for Fig. 2?

-It seems appropriate to refer in this manuscript to this relevant papers/books:

- Fornito, A., Zalesky, A., & Bullmore, E. (2016). Fundamentals of brain network analysis. Academic Press.

- Di Perri, C., Bahri, M. A., Amico, E., Thibaut, A., Heine, L., Antonopoulos, G., ... & Tshibanda, L. (2016). Neural correlates of consciousness in patients who have emerged from a minimally conscious state: a cross-sectional multimodal imaging study. The Lancet Neurology, 15(8), 830-842.

- De Vico Fallani, F., Astolfi, L., Cincotti, F., Mattia, D., la Rocca, D., Maksuti, E., ... & Nagy, Z. (2009). Evaluation of the brain network organization from EEG signals: a preliminary evidence in stroke patient. The Anatomical Record: Advances in Integrative Anatomy and Evolutionary Biology: Advances in Integrative Anatomy and Evolutionary Biology, 292(12), 2023-2031.

- Amico, E., Marinazzo, D., Di Perri, C., Heine, L., Annen, J., Martial, C., ... & Goñi, J. (2017). Mapping the functional connectome traits of levels of consciousness. NeuroImage, 148, 201-211.

- De Vico Fallani, F., Richiardi, J., Chavez, M., & Achard, S. (2014). Graph analysis of functional brain networks: practical issues in translational neuroscience. Philosophical Transactions of the Royal Society B: Biological Sciences, 369(1653), 20130521.